# [CLS] is Not Enough: Multi-Label Recognition via Patch-Level Inference and Adaptive Aggregation

**Akang Wang** [1]  **Xili Deng** [1]  **Zhanxuan Hu** [1]  **Yi Zhao** [1]  **Yonghang Tai** [1]  **Huafeng Li** [2]

## Abstract

Vision-Language Models such as CLIP exhibit strong zero-shot recognition capability by aligning images with textual concepts, yet they often underperform on multi-label recognition where multiple objects co-exist. A key bottleneck is that the [CLS] token, as a single global visual representation, is insufficient to faithfully encode diverse targets with varying scales, contexts, and co-occurrence patterns. To address this limitation, we present a new multi-label image recognition framework, termed **PIAA**, which formulates prediction as *Patch-level Inference followed by Adaptive Aggregation*. Specifically, we first enhance patch-wise predictions from two complementary perspectives: (i) mitigating semantic entanglement in the visual encoder to obtain more discriminative patch representations, and (ii) learning an unsupervised visual classifier to narrow the vision–language modality gap. We then introduce an adaptive aggregation module that consolidates patch-level scores into the final multi-label prediction. Notably, the entire pipeline is fully *training-free*, requiring no gradient updates or parameter fine-tuning. Experiments show that our method achieves strong improvements with minimal extra computation, exceeding a 6% mAP gain on the challenging NUS-WIDE benchmark over representative baselines. Code is available at https://github.com/akang-wang/PIAA.

## 1. Introduction

Over the past few years, vision-language models (VLMs) such as CLIP (Radford et al., 2021) have become general-

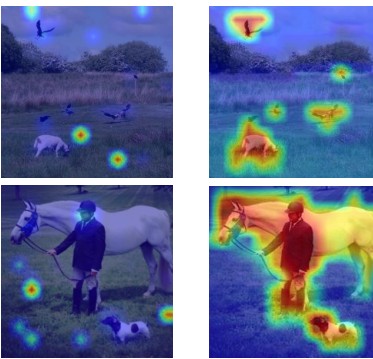

*Figure 1.* Comparison of attention and activation maps. Left: CLIP [CLS] attention is diffuse and often misses true foreground objects. Right: our learned visual classifier yields top-$K$ activation heatmaps that are more localized and object-aligned, indicating improved semantic grounding.

purpose recognition engines trained on large-scale image-text pairs. By contrastively aligning images with natural-language concepts, they enable open-vocabulary transfer and strong zero-shot recognition, and are widely adapted via prompting or lightweight tuning (Khattak et al., 2023; Wu et al., 2024). However, most VLMs still compress an image into a single global [CLS] token, implicitly assuming one dominant semantic target (Zhong et al., 2022). This global bottleneck works well for single-label recognition but often fails in multi-label scenarios with multiple co-occurring objects at diverse scales and contexts (as illustrated in Fig. 1).

To cope with multi-label recognition, a prevalent line of work reformulates it as a collection of single-label predictions by cropping the image into object-centric crops (e.g., via class activation maps (Zhou et al., 2016), attention mechanisms (Ridnik et al., 2023), or region refinement (Abdelfattah et al., 2023)) and applying a VLM to each region. While such multi-crop pipelines improve accuracy by promoting localized reasoning and reducing background interference, they come with substantial overhead: each image requires multiple forward passes, increasing computation time, memory footprint, and engineering complexity. In addition, their effectiveness is tightly coupled to crop quality and coverage, which can undermine robustness and scalability.

---

[1]Yunnan Normal University, Kunming, China [2]Kunming University of Science and Technology, Kunming, China. Correspondence to: Zhanxuan Hu <zhanxuanhu@gmail.com>.

*Proceedings of the 43rd International Conference on Machine Learning*, Seoul, South Korea. PMLR 306, 2026. Copyright 2026 by the author(s).

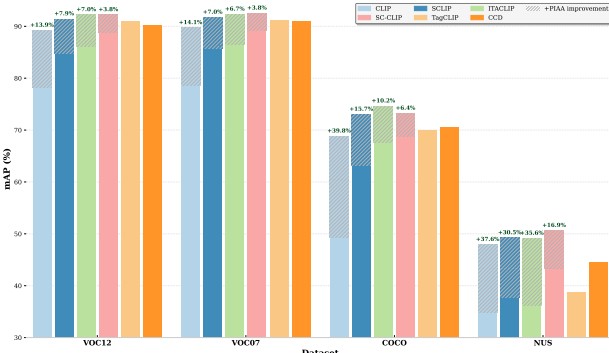

*Figure 2.* Comparison of mAP performance across four multi-label datasets. TagCLIP and CCD are representative multi-label recognition methods, while CLIP, ITACLIP, SC-CLIP, and SCLIP are originally designed for semantic segmentation. **PIAA** denotes our proposed method built upon segmentation-style inference with additional improvements.

A complementary perspective is that multi-label recognition is intrinsically a weaker task than semantic segmentation: the former only requires identifying what categories are present, whereas the latter additionally demands precise localization. This observation naturally raises the question of whether segmentation-style patch reasoning can be exploited to simplify multi-label recognition. To explore this idea, we benchmark several recent training-free semantic segmentation methods under a fair and controlled setting [1]. As shown in Fig. 2, we obtain image-level multi-label predictions by converting dense patch-level scores into per-class confidences via a simple category-wise max pooling over patches. Remarkably, CLIP-based segmentation baselines such as SC-CLIP (Bai et al., 2025) are already highly competitive with representative multi-label recognition methods (e.g., TagCLIP (Lin et al., 2024) and CCD (Kim & Shim, 2025)) across all evaluated datasets. *These results highlight the promise of addressing multi-label recognition through patch-level prediction and aggregation.*

Building on the above observations, we propose **PIAA**, a training-free framework that views multi-label recognition as patch-level inference followed by adaptive aggregation. The core of **PIAA** is to improve patch-wise predictions, and we tackle this problem from two complementary directions. First, we mitigate semantic entanglement in the visual encoder to obtain more discriminative patch representations. Semantic entanglement refers to the tendency of patch tokens to mix foreground semantics with irrelevant context or co-occurring objects, leading to ambiguous local features and unstable patch-level responses. This issue has been extensively studied in the semantic segmentation liter-

ature (Wang et al., 2024a; Aydın et al., 2025), and existing disentanglement techniques can be naturally transferred to our patch-level inference setting.

Beyond semantic entanglement, however, we emphasize another critical factor that is largely overlooked by existing training-free segmentation-style pipelines: the modality gap between the visual and textual spaces in vision–language models (Liang et al., 2022). While CLIP-like models align image-level embeddings with text, patch embeddings are often not well calibrated to text prototypes, making text-based classification at the patch level unreliable. To bridge this gap without backpropagation, we introduce a patch-based unsupervised visual classifier learning module (Zhang et al., 2025b). Specifically, leveraging abundant unlabeled images, we collect patch embeddings and learn a visual classifier directly in the visual feature space in an unsupervised manner. The learned visual classifier is then used to replace the original text classifier at inference, producing more reliable patch-level class scores without any gradient-based fine-tuning. Together, these two components yield stronger patch-wise evidence.

Further, since patch-level predictions are inevitably noisy, we introduce an adaptive aggregation mechanism to consolidate patch-wise evidence into a robust image-level multi-label prediction. By automatically down-weighting outlier patches and suppressing inconsistent activations, the proposed aggregation strategy effectively filters noise while preserving discriminative cues from informative regions. In summary, **PIAA** improves multi-label recognition by jointly enhancing patch representation quality, reducing vision–language modality discrepancy, and adaptively aggregating patch-wise predictions, enabling accurate and efficient multi-label inference with a single forward pass and without backpropagation. The main contributions are summarized as follows:

- *A new perspective.* We revisit multi-label recognition with vision–language models from a patch-level viewpoint, shifting the prediction paradigm from a single global [CLS] representation to fine-grained patch-wise inference.

- *A simple yet effective framework.* We propose **PIAA**, a training-free framework that improves patch-level predictions by mitigating semantic entanglement and reducing the vision–language modality gap via unsupervised visual classifier learning, followed by adaptive aggregation for robust image-level inference.

- *Promising results.* Extensive experiments on challenging multi-label benchmarks demonstrate that **PIAA** consistently outperforms representative baselines by a clear margin, achieving substantial performance gains with minimal extra computation.

---

[1]Specifically, we restrict our comparison to CLIP-based approaches and exclude methods that rely on additional vision backbones or external segmenters (e.g., DINO or SAM-based models).

## 2. Related Work

### 2.1. Multi-Label Classification.

Multi-Label Image Classification (MLC) aims to recognize multiple co-existing semantic entities within a single image. Early research predominantly focused on fully supervised learning, where methods such as BCE-LS (Cole et al., 2021) rely on exhaustive multi-label annotations to model label dependencies and achieve strong performance, albeit at the cost of expensive annotation and limited scalability. With the advent of vision-language models (VLMs), the reliance on dense supervision has been substantially reduced, giving rise to a growing body of weakly supervised (Pu et al., 2022; Chen et al., 2022), unsupervised (Kim & Shim, 2025; Abdelfattah et al., 2023), and zero-shot (Hu et al., 2026; Liu et al., 2024; Zhang et al., 2024; Lin et al., 2024; Miller et al., 2025) methods.

Weakly supervised methods like single-positive learning (Chen et al., 2024; Xing et al., 2024; Tran et al., 2025), and parameter-efficient visual prompt tuning (e.g., ML-VPT (Ma et al., 2025)) achieve strong performance but fundamentally rely on task-specific annotations and gradient-based training. Meanwhile, unsupervised methods such as CDUL (Abdelfattah et al., 2023) and CCD (Kim & Shim, 2025) alleviate label dependence via pseudo-labeling and self-distillation, yet their iterative optimization introduces substantial computational overhead. In contrast to these training-heavy paradigms, we draw inspiration from training-free segmentation-style inference (e.g., SPARC (Miller et al., 2025)) and propose **PIAA**. Our method demonstrates that effective patch-level prediction and adaptive aggregation offer a simple, optimization-free alternative for efficient multi-label recognition.

### 2.2. Open-vocabulary Semantic Segmentation.

Open-vocabulary semantic segmentation (OVSS) has recently attracted increasing attention as it enables training-free dense labeling by repurposing vision–language foundation models such as CLIP. At its core, OVSS aims to alleviate CLIP's intrinsic limitations for dense prediction: CLIP is primarily optimized for global image–text alignment, whereas segmentation requires localized patch-wise semantics and reliable separation between foreground and background. Existing OVSS methods can be broadly grouped into two lines. *(i) CLIP-only adaptation* modifies CLIP's internal mechanisms to better preserve local cues and suppress background interference, e.g., repurposing attention for dense masks (Dong et al., 2023), enhancing intra-object grouping via correlation-aware attention (Wang et al., 2024a), masking distracting regions or fusing multi-layer features (Lin et al., 2024; Aydın et al., 2025), and calibrating dense outputs by neutralizing anomaly tokens (Bai et al., 2025). *(ii) External-prior based methods* introduce

structural priors from additional foundation models to refine localization, such as enforcing geometric/spatial consistency with VFMs (Lan et al., 2024; Zhang et al., 2025a), leveraging SAM-style mask proposals, or exploiting diffusion cross-attention as implicit localization cues (Wang et al., 2025; Zhou et al., 2025).

While external-prior-based approaches can yield stronger masks, they inevitably introduce additional models, leading to extra forward passes, higher memory footprint, and increased system complexity. In contrast, CLIP-only methods are lightweight and have achieved strong results by refining attention and feature aggregation. However, they predominantly focus on improving the quality of patch-level predictions while overlooking the modality discrepancy between localized visual evidence and global text prototypes. Moreover, how to reliably aggregate patch-wise scores into robust image-level outputs remains under-explored.

## 3. Method

### 3.1. Preliminaries

**VLM-based Multi-Label Recognition.** Given an image $I$, a vision–language model (e.g., CLIP) encodes it into a global visual embedding $\mathbf{z}_{\mathrm{cls}} \in \mathbb{R}^d$ and aligns it with a set of textual prototypes $\{\mathbf{w}_c\}_{c=1}^C$ constructed from category names. Standard zero-shot prediction is obtained by computing the cosine similarity between $\mathbf{z}_{\mathrm{cls}}$ and each $\mathbf{w}_c$, followed by thresholding to produce multiple labels. Despite its simplicity and efficiency, this paradigm is inherently suboptimal for multi-label recognition. First, the global `[CLS]` token $\mathbf{z}_{\mathrm{cls}}$ is optimized to capture the dominant semantics of an image, and thus often overlooks multiple co-existing objects, particularly those that are small, occluded, or less salient. Second, existing attempts to mitigate these issues typically resort to heuristic region selection or multi-crop inference, which introduces additional computational overhead. These limitations motivate us to move beyond `[CLS]`-based reasoning and develop a training-free framework that enables reliable *patch-level inference* and *adaptive aggregation*.

### 3.2. Our Method

**Overview.** To overcome the limitations of the global `[CLS]` token in multi-label scenarios, we propose **PIAA** (Fig. 3), a training-free framework that reformulates recognition as *patch-level inference* followed by *adaptive aggregation*. A key advantage of **PIAA** is its modularity: the visual extraction process can be equipped with *any* CLIP-based segmentation-style variant or disentanglement front-end (e.g., ITACLIP (Aydın et al., 2025), SC-CLIP (Bai et al., 2025)), which provides dense, discriminative patch representations by suppressing background interference and improving locality. Building on these patch features, **PIAA**

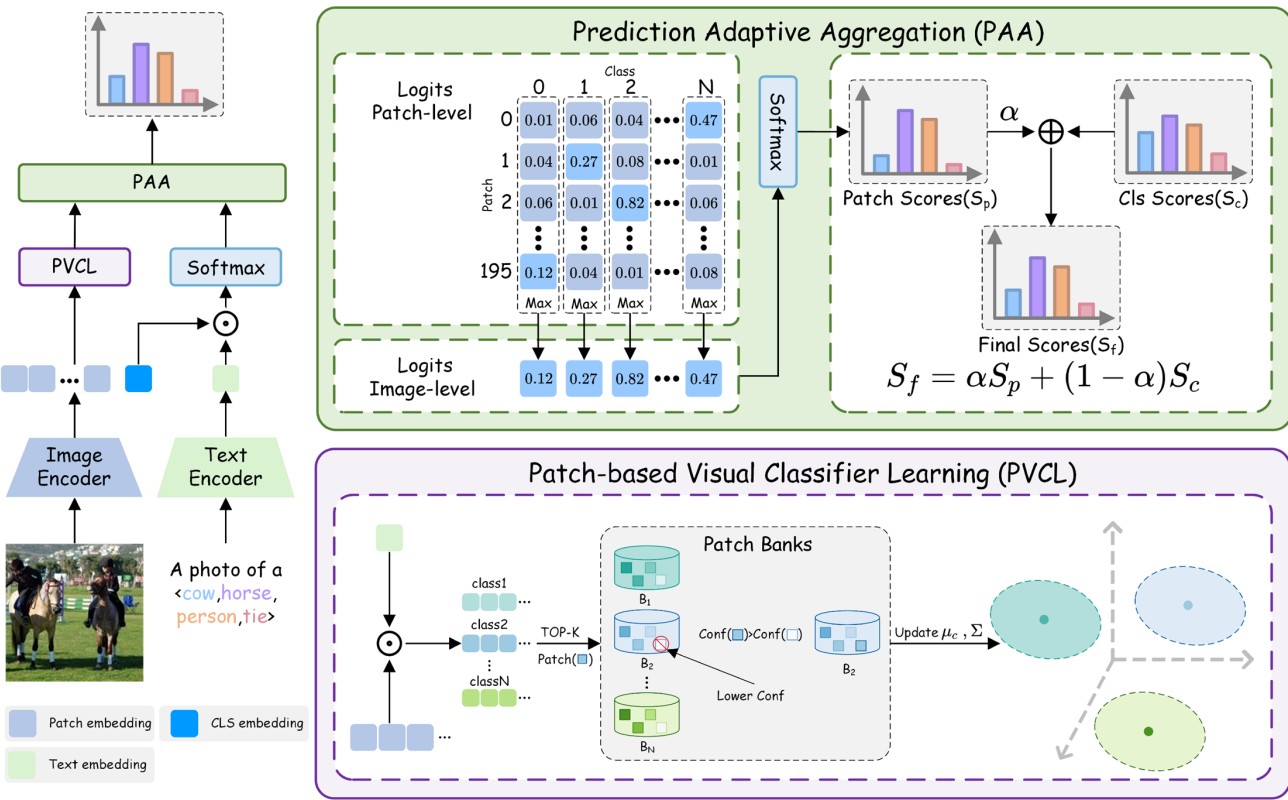

*Figure 3.* Overview of the proposed **PIAA** framework. Given an input image, a CLIP-based segmentation-style image encoder (optionally enhanced by semantic disentanglement) produces patch embeddings and a global [CLS] embedding. **PIAA** consists of two components. **(i) PVCL** learns a patch-based visual classifier from the patch embeddings, aiming to reduce the vision–language modality gap during inference and improve patch-level scores. **(ii) PAA** adaptively combines the calibrated patch-level predictions with the [CLS]-based predictions to produce final image-level multi-label outputs. The overall pipeline is training-free and requires no backpropagation.

improves multi-label inference via two synergistic components. *(1) Patch-based Visual Classifier Learning (PVCL)*: it rectifies the vision–language modality gap by analytically estimating an unsupervised visual classifier directly from the patch representations. *(2) Prediction Adaptive Aggregation (PAA)*: it consolidates localized evidence into robust image-level predictions by dynamically fusing fine-grained patch scores with global [CLS] semantic anchors.

### 3.3. Patch-based Visual Classifier Learning (PVCL)

A primary bottleneck in VLM-based multi-label recognition lies in the *vision–language modality gap* (Liang et al., 2022; Huang et al., 2025): while textual prototypes provide robust global semantics, they often struggle to faithfully represent localized patch features, leading to unreliable fine-grained predictions. An elegant strategy to bridge this gap is to *learn an unsupervised visual classifier directly within the visual embedding space* (Zanella et al., 2024; Zhang et al., 2025b). By deriving the classifier purely from visual feature statistics, this approach inherently circumvents cross-modal misalignment while completely eliminating the need for gradient-based training. A representative instantiation

of this concept is Gaussian Discriminant Analysis (GDA), which admits a *closed-form* solution for classifier parameters and has recently proven highly effective for training-free classifier rectification (Hastie & Tibshirani, 1996; Wang et al., 2024b; Zhu et al., 2024).

Formally, let $\mathcal{X} = \{\mathbf{x}_i\}_{i=1}^M$ denote the collection of unlabeled patch features for a $C$-way classification problem. To achieve efficient, closed-form adaptation, we assume that the patch features conditioned on class $c$ follow a multivariate Gaussian distribution with a shared covariance matrix:

$$
\begin{aligned}
p_{i,c} &= p(\mathbf{x}_i \mid y = c) = \mathcal{N}(\mathbf{x}_i; \boldsymbol{\mu}_c, \boldsymbol{\Sigma}) \\
&= \frac{1}{\sqrt{(2\pi)^d |\boldsymbol{\Sigma}|}} \exp\left( -\frac{1}{2} (\mathbf{x}_i - \boldsymbol{\mu}_c)^\top \boldsymbol{\Sigma}^{-1} (\mathbf{x}_i - \boldsymbol{\mu}_c) \right),
\end{aligned}
\tag{1}
$$

where $d$ is the feature dimension, $\boldsymbol{\mu}_c$ is the class mean, and $\boldsymbol{\Sigma}$ is the shared covariance matrix. While this assumption may simplify the real-world feature distributions, it crucially enables analytical inference with minimal computational overhead, eliminating the need for gradient-based optimization.

According to Bayes' theorem, the posterior probability of class $c$ given the $i$-th patch feature $\mathbf{x}_i$ is given by:

$$p(y = c \mid \mathbf{x}_i) \propto \pi_c \, \mathcal{N}(\mathbf{x}_i; \boldsymbol{\mu}_c, \boldsymbol{\Sigma}), \tag{2}$$

where $\pi_c$ is the class prior. Assuming a uniform class prior $\pi_c = 1/C$, the Bayes optimal prediction simplifies to maximizing the log-posterior. By expanding the Gaussian density function and discarding terms independent of $c$, the decision boundary becomes strictly linear. Thus, the discriminant score for class $c$ is:

$$\tilde{y}_{i,c} = \mathbf{w}_c^\top \mathbf{x}_i + b_c, \tag{3}$$

where the closed-form classifier weights $\mathbf{w}_c$ and biases $b_c$ are given by:

$$\mathbf{w}_c = \boldsymbol{\Sigma}^{-1} \boldsymbol{\mu}_c, \qquad b_c = -\frac{1}{2} \boldsymbol{\mu}_c^\top \boldsymbol{\Sigma}^{-1} \boldsymbol{\mu}_c. \tag{4}$$

Under the standard i.i.d. assumption, the maximum likelihood estimators for $\boldsymbol{\mu}_c$ and $\boldsymbol{\Sigma}$ simply correspond to the class-wise empirical averages and the pooled sample covariance matrix of $\mathcal{X}$. However, applying GDA directly to the raw patch manifold is highly vulnerable to background clutter and semantic ambiguity in unsupervised scenarios. To address this, we propose a three-stage purification algorithm that transitions from initial cross-modal bootstrapping to refined intra-modal estimation.

*Stage I: Entropy-Guided Bootstrapping.* To harvest reliable anchors while filtering background noise, we leverage the zero-shot text-alignment probability $p_{i,c}$ representing the likelihood of patch $\mathbf{x}_i$ belonging to class $c$ to identify patches with minimum predictive entropy:

$$H(\mathbf{x}_i) = -\sum_{c=1}^{C} p_{i,c} \log p_{i,c}. \tag{5}$$

For each class $c$, we initialize a pristine memory bank $\mathcal{B}_c^{(0)}$ by harvesting the top-$K$ patches with the lowest entropy:

$$\mathcal{B}_c^{(0)} = \left\{ \mathbf{x}_i \in \mathcal{X} \;\middle|\; \arg\max_k p_{i,k} = c \right\}_{\text{Top-}K \text{ by } \min H(\mathbf{x}_i)}. \tag{6}$$

Using this initial bank, we compute the preliminary empirical means $\boldsymbol{\mu}_c^{(0)}$ and a pooled covariance $\boldsymbol{\Sigma}^{(0)}$ to instantiate a temporary GDA classifier.

*Stage II: Vision-Driven Purification.* To bridge the vision-language modality gap and its associated artifacts in $\mathcal{B}_c^{(0)}$, we re-evaluate the bank using the preliminary GDA classifier to obtain purely *vision-driven* scores $q_{i,c}$. By assuming a Gaussian distribution of these scores, we strictly purify the bank using an adaptive statistical threshold based on their empirical mean $\mu_{q,c}$ and standard deviation $\sigma_{q,c}$:

$$\mathcal{B}_c = \left\{ \mathbf{x}_i \in \mathcal{B}_c^{(0)} \;\middle|\; q_{i,c} \geq \mu_{q,c} + \sigma_{q,c} \right\}. \tag{7}$$

---

**Algorithm 1** Patch-based Visual Classifier Learning(PVCL)

---

1: **Input:** Unlabeled patch manifold $\mathcal{X}$, bank size $K$.
2: **Output:** Final classifier weights $\{\mathbf{w}_c, b_c\}_{c=1}^{C}$.
3: /* *Stage I: Entropy-Guided Bootstrapping* */
4: Initialize $\mathcal{B}_c^{(0)} \leftarrow$ top-$K$ patches in $\mathcal{X}$ minimizing $H(\mathbf{x}_i) = -\sum_c p_{i,c} \log p_{i,c}$ for all $c$.
5: Estimate preliminary means $\boldsymbol{\mu}_c^{(0)}$ and covariance $\boldsymbol{\Sigma}^{(0)}$ via $\mathcal{B}^{(0)}$ to form temporary classifier $\{\mathbf{w}_c^{(0)}, b_c^{(0)}\}$.
6: /* *Stage II: Vision-Driven Purification* */
7: Compute vision-driven scores $q_{i,c} \leftarrow$ Softmax$(\mathbf{w}_c^{(0)\top} \mathbf{x}_i + b_c^{(0)})$ for all $\mathbf{x}_i \in \mathcal{B}_c^{(0)}$.
8: Purify bank: $\mathcal{B}_c \leftarrow \left\{ \mathbf{x}_i \in \mathcal{B}_c^{(0)} \;\middle|\; q_{i,c} \geq \mu_{q,c} + \sigma_{q,c} \right\}$ using class-wise empirical statistics.
9: /* *Stage III: Robust Shrinkage Induction* */
10: Compute confidence-weighted prototypes $\boldsymbol{\mu}_c$ and pooled empirical covariance $\hat{\boldsymbol{\Sigma}}$ via $\mathcal{B}$.
11: Apply trace-regularized shrinkage: $\boldsymbol{\Sigma}^{-1} \leftarrow d\left[(|\mathcal{B}| - 1)\hat{\boldsymbol{\Sigma}} + \text{Tr}(\hat{\boldsymbol{\Sigma}})\mathbf{I}_d\right]^{-1}$.
12: Derive final weights: $\mathbf{w}_c \leftarrow \boldsymbol{\Sigma}^{-1} \boldsymbol{\mu}_c$ and biases $b_c \leftarrow -\frac{1}{2}\boldsymbol{\mu}_c^\top \boldsymbol{\Sigma}^{-1} \boldsymbol{\mu}_c$.
13: **return** $\{\mathbf{w}_c, b_c\}_{c=1}^{C}$

---

*Stage III: Robust Shrinkage Induction.* Armed with the strictly purified visual bank $\mathcal{B}$, we compute the final GDA parameters. To further mitigate residual noise, the class prototypes $\boldsymbol{\mu}_c$ are computed as confidence-weighted averages. We leverage the vision-driven probabilities $q_{i,c}$ derived from the Stage II temporary classifier to avoid the modality gap:

$$\boldsymbol{\mu}_c = \frac{\sum_{\mathbf{x}_i \in \mathcal{B}_c} q_{i,c} \mathbf{x}_i}{\sum_{\mathbf{x}_i \in \mathcal{B}_c} q_{i,c}}. \tag{8}$$

Subsequently, we compute the pooled sample covariance matrix $\hat{\boldsymbol{\Sigma}}$ across all classes:

$$\hat{\boldsymbol{\Sigma}} = \frac{1}{|\mathcal{B}|} \sum_{c=1}^{C} \sum_{\mathbf{x}_i \in \mathcal{B}_c} (\mathbf{x}_i - \boldsymbol{\mu}_c)(\mathbf{x}_i - \boldsymbol{\mu}_c)^\top. \tag{9}$$

Since estimating the precision matrix in high-dimensional embedding spaces (e.g., $d \geq 512$) is often ill-posed and numerically unstable, we apply a trace-regularized shrinkage estimator:

$$\boldsymbol{\Sigma}^{-1} = d\left[(|\mathcal{B}| - 1)\hat{\boldsymbol{\Sigma}} + \text{Tr}(\hat{\boldsymbol{\Sigma}})\mathbf{I}_d\right]^{-1}. \tag{10}$$

Finally, the optimal, backpropagation-free visual classifier weights $\{\mathbf{w}_c, b_c\}_{c=1}^{C}$ are analytically derived using Eq. (4). By progressively shifting from textual priors to visual distributions, this process yields highly discriminative patch-level predictors. The complete procedure of PVCL is summarized in Algorithm 1.

## 3.4. Prediction Adaptive Aggregation (PAA)

Building upon the rectified patch-level scores from PVCL, the remaining challenge is to effectively aggregate these localized responses into a reliable image-level prediction. While PVCL significantly enhances the semantic validity of individual patches, dense patch-wise predictions inevitably contain residual noise from background clutter. Given the spatial sparsity inherent to multi-label recognition, where target objects often occupy only a small fraction of the image, a naive global average pooling would inadvertently dilute the localized signals.

*Spatial Evidence Distillation.* Recall that the calibrated discriminant score for the $i$-th patch regarding class $c$ is $\tilde{y}_{i,c} = \mathbf{w}_c^\top \mathbf{x}_i + b_c$. Since these GDA outputs are unbounded logits, we first normalize them into patch-wise probabilities:

$$p_{i,c} = \frac{\exp(\tilde{y}_{i,c})}{\sum_{k=1}^{C} \exp(\tilde{y}_{i,k})}. \tag{11}$$

Next, to isolate the most discriminative localized evidence, we perform category-wise max-pooling over these probabilities. However, these independently aggregated peak scores ($\max_i p_{i,c}$) do not naturally sum to one. To form a valid image-level categorical distribution for the subsequent global-local fusion, we directly apply a secondary Softmax recalibration:

$$S_{\text{patch},c} = \text{Softmax} \left( \max_{i \in \{1, \dots, M\}} p_{i,c} \right). \tag{12}$$

*Global-Local Adaptive Fusion.* Crucially, as supported by our empirical analysis detailed in Appendix A.3, we observe a fundamental complementarity between local and global semantic representations based on object scale and saliency. While max-aggregated patch evidence ($S_{\text{patch},c}$) excels at identifying small or localized objects, it inherently lacks a macroscopic perspective, making it occasionally unstable for large or globally salient objects that span multiple patches. Conversely, the global [CLS] token softmax prediction, denoted as $S_{\text{cls},c}$, acts as a holistic semantic anchor that inherently captures large-scale, scene-level consistency.

To exploit both local sensitivity and global semantic stability, we formulate the final prediction $S_{f,c}$ as a convex combination of the two complementary representations:

$$S_{f,c} = \alpha S_{\text{patch},c} + (1 - \alpha) S_{\text{cls},c}, \tag{13}$$

where the coefficient $\alpha \in [0, 1]$ controls the trade-off. In practice, assigning a decisively larger weight to the patch-level evidence (e.g., $\alpha = 0.9$) preserves the fine-grained localization capabilities essential for small objects, while the [CLS]-based prediction effectively regularizes the outputs for large, salient targets. This dual-path adaptive aggregation ultimately yields substantially larger performance gains than using either component alone.

## 4. Experiments

### 4.1. Experimental Setup

**Datasets.** We evaluate **PIAA** on four diverse multi-label benchmarks. PASCAL VOC 2007 and 2012 (Everingham et al., 2010) (20 classes) serve as foundational benchmarks, containing 5,011/4,952 and 5,717/5,823 images for their respective train/test splits. To assess performance in complex scenes, we utilize MS COCO (Lin et al., 2014), which comprises 80 categories with 82,081 training and 40,137 validation images. Finally, we scale the evaluation to NUS-WIDE (Chua et al., 2009), encompassing 81 concepts across 150,000 training and 60,260 test samples. These datasets provide a broad spectrum of label densities and object scales, facilitating a robust assessment of our approach.

**Evaluation Metrics.** Following standard protocols (Everingham et al., 2010), we employ mean Average Precision (mAP) as the primary metric. Crucially, our framework operates in a strictly training-free and unsupervised manner (Kim & Shim, 2025; Abdelfattah et al., 2023): ground-truth labels are sequestered solely for performance evaluation and remain entirely inaccessible during patch bank construction and inference. This ensures the integrity of our zero-shot transfer paradigm.

**Implementation Details.** **PIAA** is instantiated with a frozen CLIP ViT-B/16 backbone, operating strictly without gradient updates. For PVCL, class-specific memory banks are curated by selecting the top $K = 512$ patches from the unlabeled manifold based on minimal predictive Shannon entropy, from which the closed-form classifiers are analytically derived. We uniformly set $K = 512$ across all benchmarks, validating the robustness of our entropy-driven selection without per-dataset tuning. During inference, PIAA fuses the aggregated patch scores with the global [CLS] scores using a trade-off parameter $\alpha = 0.9$. This heavily weights localized discriminative cues to uncover co-existing objects while retaining global context for regularization.

**Baselines.** We comprehensively benchmark **PIAA** across diverse supervision paradigms: *(1) Supervised & Weakly Supervised:* We include BCE-LS (Cole et al., 2021) as a fully supervised upper bound, alongside robust single-positive learning methods (VLPL (Xing et al., 2024), AEVLP (Tran et al., 2025)). *(2) Unsupervised (Training-based):* We evaluate CDUL (Abdelfattah et al., 2023) and CCD (Kim & Shim, 2025), which rely on iterative pseudo-labeling and self-distillation. *(3) Training-free (Ours):* We compare against vanilla CLIP (Radford et al., 2021) and recent adaptations like TagCLIP (Lin et al., 2024) (heuristic token masking) and SPARC (Miller et al., 2025) (score prompting and adaptive fusion). Unlike CCD, which requires iterative optimization, or SPARC, which relies on constructing com-

*Table 1.* Comparison of mean Average Precision (mAP, %) on four multi-label benchmarks. We evaluate **PIAA** against various supervision paradigms. **PIAA** consistently establishes a new state-of-the-art (SOTA) among training-free and unsupervised methods, even rivaling fully supervised baselines. Bold indicates the best performance in training-free and unsupervised settings.

| Supervision level | Annotation | Method | Frozen | VOC12 | VOC07 | COCO | NUS |
|---|---|---|---|---|---|---|---|
| Fully supervised | Fully labeled | BCE-LS (Cole et al., 2021) | × | 91.6 | 92.6 | 79.4 | 51.7 |
| Weakly supervised | Partial labeled (10%) | ASL (Ridnik et al., 2021) | × | - | 82.9 | 69.7 | - |
| | | SARB (Pu et al., 2022) | × | - | 85.7 | 72.5 | - |
| | | Chen et al. (Chen et al., 2022) | × | - | 81.5 | 68.1 | - |
| | Single positive labeled | LL-R (Kim et al., 2022) | × | 89.7 | 90.6 | 72.6 | 47.4 |
| | | $G^2$ NetPL (Abdelfattah et al., 2022) | × | 89.5 | 89.9 | 72.5 | 48.5 |
| | | GR Loss (Chen et al., 2024) | × | 89.8 | - | 73.2 | 49.1 |
| | | VLPL (Xing et al., 2024) | × | 89.1 | - | 71.5 | 49.6 |
| | | AEVLP (Tran et al., 2025) | × | 90.5 | - | 73.5 | 50.7 |
| Unsupervised | Annotation free | NaiveAN (Kundu & Tighe, 2020) | × | 85.5 | 86.5 | 65.1 | 40.8 |
| | | ROLE (Cole et al., 2021) | × | 82.6 | 84.6 | 67.1 | 43.2 |
| | | CDUL (Abdelfattah et al., 2023) | × | 88.6 | 89.0 | 69.2 | 44.0 |
| | | CCD (Kim & Shim, 2025) | × | 90.1 | 91.0 | 70.3 | 44.5 |
| Training free | Annotation free | CLIP (Radford et al., 2021) | ✓ | 84.9 | 85.4 | 61.7 | 44.4 |
| | | TagCLIP (Lin et al., 2024) | ✓ | 90.8 | 91.2 | 70.0 | 38.7 |
| | | SPARC (Miller et al., 2025) | ✓ | - | 88.7 | 68.0 | 47.5 |
| | | PIAA (ours) | ✓ | **92.2** | **92.5** | **73.2** | **50.6** |

plex compound prompts and aggregating multiple inference scores, **PIAA** establishes a new state-of-the-art in a purely training-free, single-pass manner by analytically modeling patch-level visual distributions.

### 4.2. Main Results

**Overall Results.** As reported in Tab. 1, **PIAA** consistently achieves the best performance across all four benchmarks among training-free and unsupervised methods, and is competitive with weakly supervised and even fully supervised approaches. Compared with the original CLIP baseline, recent unsupervised methods such as CDUL and CCD improve performance by leveraging unlabeled data to refine classifiers and representations, while training-free methods such as TagCLIP and SPARC further boost performance through inference-time calibration and spatial reasoning.

Building on these insights, **PIAA** establishes a new state of the art by jointly addressing two key limitations: the vision–language modality gap and the lack of principled patch-to-image aggregation. On the challenging NUS-WIDE dataset, **PIAA** surpasses the strongest training-free method (TagCLIP) by **11.9%** mAP and outperforms the optimization-based CCD by **6.1%** mAP, despite requiring no training, backpropagation, or manual annotations. Moreover, **PIAA** achieves performance comparable to fully supervised methods on VOC2007 and VOC2012, and rivals strong weakly supervised methods on MS COCO. These results demonstrate that fine-grained, training-free patch-level reasoning can be more effective and robust than traditional training-based unsupervised paradigms.

*Table 2.* Ablation Study of **PIAA** Components. Performance (mAP, %) is evaluated on four benchmarks using SC-CLIP as the disentanglement front-end. ✓ indicates the activation of a module.

| PVCL | PAA | VOC12 | VOC07 | COCO | NUS |
|---|---|---|---|---|---|
| | ✓ | 88.8 | 89.1 | 68.8 | 43.3 |
| | ✓ | 89.6 | 90.4 | 70.3 | 45.3 |
| ✓ | | 91.3 | 91.7 | 69.9 | 45.7 |
| ✓ | ✓ | **92.2** | **92.5** | **73.2** | **50.6** |

### 4.3. Ablation Study

To evaluate the individual contributions and joint synergy of the proposed components within the **PIAA** framework, we conduct a series of systematic ablation studies. As detailed in Tab. 2, Tab. 3, and Tab. 4, our empirical results highlight the progressive performance enhancements and the robust compatibility among various vision backbones, disentanglement techniques, PVCL, and PAA.

**Effect of PVCL.** As demonstrated in Tab. 2, integrating PVCL standalone consistently enhances performance across all benchmarks. Specifically, it yields a +2.6% mAP gain on VOC07 (89.1% to 91.7%) and a +2.4% improvement on the challenging NUS-WIDE dataset (43.3% to 45.7%). This demonstrates that even with spatially purified features, the vision-language modality gap remains a bottleneck. PVCL successfully addresses this by rectifying decision boundaries within the visual manifold. By substituting uncalibrated textual prototypes with statistically-estimated classifiers, it ensures that patch-level scores are strictly grounded in visual evidence, delivering more robust fine-grained predictions.

*Table 3.* Performance gains of **PIAA** across different segmentation-style variants based on the ViT-B/16 architecture. While various attention disentanglement techniques yield superior base results over vanilla CLIP, **PIAA** provides significant and orthogonal improvements to all evaluated front-ends, achieving an average mAP surge of +13.7% even on the standard CLIP baseline.

| Method | VOC12 | VOC07 | COCO | NUS | Average |
|---|---|---|---|---|---|
| CLIP | 78.3 | 78.6 | 49.2 | 34.8 | 60.2 |
| + PIAA | $89.2_{+10.9}$ | $89.7_{+11.1}$ | $68.8_{+19.6}$ | $47.9_{+13.1}$ | $73.9_{+13.7}$ |
| SCLIP | 84.7 | 85.7 | 63.1 | 37.7 | 67.8 |
| + PIAA | $91.4_{+6.7}$ | $91.7_{+6.0}$ | $73.0_{+9.9}$ | $49.2_{+11.5}$ | $76.3_{+8.5}$ |
| ITACLIP | 86.2 | 86.5 | 67.7 | 36.2 | 69.2 |
| + PIAA | $92.2_{+6.0}$ | $92.3_{+5.8}$ | $74.6_{+6.9}$ | $49.1_{+12.9}$ | $77.1_{+7.9}$ |
| SC-CLIP | 88.8 | 89.1 | 68.8 | 43.3 | 72.5 |
| + PIAA | $92.2_{+3.4}$ | $92.5_{+3.4}$ | $73.2_{+4.4}$ | $50.6_{+7.3}$ | $77.1_{+4.6}$ |

*Table 4.* Scalability across advanced Vision Foundation Models. By including our primary baseline (ViT-B/16) as a reference, we demonstrate that **PIAA** consistently delivers substantial improvements regardless of the model scale (ViT-L) or pre-training paradigm (EVA-02).

| Architecture | Method | VOC12 | VOC07 | COCO | NUS | Average |
|---|---|---|---|---|---|---|
| *Standard CLIP* | | | | | | |
| ViT-B/16 | Base | 78.3 | 78.6 | 49.2 | 34.8 | 60.2 |
| | **+ PIAA** | 89.2 | 89.7 | 68.8 | 47.9 | 73.9 |
| ViT-L/14 | Base | 70.5 | 69.9 | 45.8 | 32.5 | 54.7 |
| | **+ PIAA** | 88.8 | 88.3 | 69.9 | 47.8 | 73.7 |
| *Advanced Foundation Model (EVA-02-CLIP)* | | | | | | |
| ViT-B/16 | Base | 83.4 | 84.0 | 60.5 | 38.3 | 66.6 |
| | **+ PIAA** | 90.0 | 90.3 | 72.2 | 46.8 | 74.8 |
| ViT-L/14 | Base | 85.5 | 83.9 | 73.3 | 39.0 | 70.4 |
| | **+ PIAA** | 93.4 | 92.9 | 79.6 | 49.4 | 78.8 |

**Effect of PAA.** The Prediction Adaptive Aggregation (PAA) module is the cornerstone for fusing localized evidence. As evidenced in Tab. 2, PAA exhibits a profound synergy with PVCL. For instance, on the complex MS-COCO dataset, combining both modules yields a +4.4% surge (reaching 73.2%), which far exceeds the sum of their individual gains (+1.1% and +1.5%, respectively). Similarly, on NUS-WIDE, adding PAA to the PVCL baseline triggers a massive +4.9% leap (from 45.7% to 50.6%). This strictly underscores that PAA reaches peak efficacy when aggregating evidence already calibrated by PVCL. By harmonizing fine-grained patch scores with the global `[CLS]` semantic anchor, PAA effectively suppresses spurious background noise, culminating in a highly robust, training-free inference pipeline.

**Synergy with Disentangled Representations.** As shown in Tab. 3, disentanglement techniques inherently improve baseline performance by modifying the attention mechanisms within the fixed ViT-B/16 architecture, with SC-CLIP achieving a strong 72.5% average mAP. Crucially, our **PIAA** acts as a powerful orthogonal enhancement across all these segmentation-style variants. Remarkably, applying PIAA to the vanilla CLIP yields a +13.7% average mAP surge. On MS-COCO, PIAA alone boosts vanilla CLIP from 49.2% to 68.8% (+19.6%), effectively matching the sophisticated SC-CLIP base model without any internal architectural modifications. Furthermore, integrating PIAA with top-performing variants like SC-CLIP pushes the overall upper bound to a formidable 77.1% average mAP. This demonstrates that PIAA successfully amplifies localized discriminative cues regardless of the underlying attention strategy.

### 4.4. Further Analyses

**Scalability to Advanced Foundation Models.** To demonstrate that our method does not merely overfit to a specific backbone, we evaluate the scalability of **PIAA** across larger architectures and more advanced Vision Foundation Models

(VFMs). As detailed in Tab. 4, we extend our analysis from the primary baseline (standard CLIP ViT-B/16) to the larger ViT-L/14, as well as the highly optimized EVA-02-CLIP family.

The results reveal a compelling and consistent scaling trajectory. For the standard CLIP ViT-L/14, which typically suffers from severe uncalibrated domain shifts in zero-shot multi-label settings (yielding a low 54.7% average mAP), integrating PIAA triggers an unprecedented surge of +19.0%, effectively rescuing the baseline. More remarkably, when applied to the state-of-the-art EVA-02 pre-training paradigm, PIAA continues to push the performance upper bound without suffering from diminishing returns. Equipping the massive EVA-02-CLIP (L14) with our module establishes a formidable new baseline, achieving a 78.8% average mAP. Crucially, on the highly complex MS-COCO dataset, this configuration approaches the 80% milestone (reaching 79.6% mAP) in a purely training-free manner. These improvements unequivocally prove that PIAA effectively harnesses the richer, high-dimensional representations of advanced VFMs, maintaining strong positive synergy regardless of the backbone's scale or pre-training strategy.

**Effect of Bank Size $K$.** The Top-$K$ selection strategy serves as a critical mechanism for distilling a pristine visual manifold while ensuring computational parsimony. By prioritizing patches with minimal predictive entropy, this approach effectively preserves class-specific feature distributions while aggressively filtering out background noise and semantically ambiguous artifacts. As illustrated in the left panel of Fig. 4, we systematically vary the bank size $K$ from 128 to 1536. Across all four benchmarks, performance consistently improves as $K$ increases up to 512, indicating that a sufficient volume of high-confidence patches is necessary to capture intra-class variance. However, relaxing the selection criteria beyond $K = 512$ leads to a gradual performance degradation. This decline elegantly confirms our

motivation: incorporating an excessive number of patches inevitably introduces high-entropy, out-of-distribution noise that corrupts the statistically-estimated classifiers. Consequently, $K = 512$ emerges as the optimal configuration, striking a perfect balance between representation richness and manifold purity.

**Analysis of Global-Local Fusion Weight $\alpha$.** The right panel of Fig. 4 evaluates the sensitivity of the hyperparameter $\alpha$, which governs the integration of fine-grained patch-level evidence and the global [CLS] semantic prior. We observe a remarkably consistent trajectory across all datasets: the performance steadily climbs as $\alpha$ increases, peaking optimally at $\alpha = 0.9$. This pronounced dominance of $\alpha$ validates our core hypothesis that localized, disentangled representations are the primary driving force for identifying co-existing targets in complex scenes. Notably, setting $\alpha = 1.0$ (which entirely discards the global token) triggers a distinct performance drop. This underscores the indispensable role of the global semantic anchor; while local patches are crucial for target localization, holistic scene context acts as a vital regularizer to suppress spurious activations and contextual false positives.

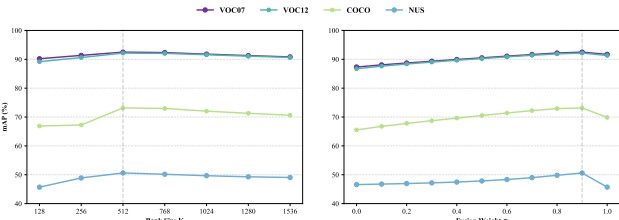

*Figure 4.* Sensitivity analysis of the patch bank capacity $K$ and the global-local fusion weight $\alpha$.

**Efficiency Analysis.** As detailed in Tab. 5, **PIAA** fundamentally redefines the efficiency-accuracy frontier by dismantling the computational bottlenecks of traditional paradigms. In terms of learning, it achieves a staggering $362.1\times$ speedup over CCD. While CCD relies on computationally exhaustive recursive self-training with redundant forward passes, **PIAA** employs a single-pass statistical sweep to derive optimal weights in closed form, slashing the adaptation time on NUS-WIDE from 991.8 to a mere 2.5 minutes. During inference, it delivers a robust $50.4\times$ average throughput boost over TagCLIP. TagCLIP suffers from sequential re-verification latency, where complexity scales linearly with vocabulary size ($\mathcal{O}(C)$). In stark contrast, **PIAA** utilizes a fully parallelized, unified forward architecture ($\mathcal{O}(1)$) that completely decouples inference speed from scene complexity. This structural elegance yields up to a $60.1\times$ advantage on densely populated datasets like MS-COCO, establishing a new standard for real-time, high-capacity multi-label recognition.

*Table 5.* Efficiency Analysis: Classifier Acquisition (Learning) and Inference (Test) Time (min).

| Dataset | Acquisition (Learning) | | | Inference (Test) | | |
|---|---|---|---|---|---|---|
| | CCD | Ours | Speedup | TagCLIP | Ours | Speedup |
| VOC12 | 43.2 | 0.2 | 216.0× | 5.8 | 0.2 | 29.0× |
| VOC07 | 45.6 | 0.2 | 228.0× | 5.4 | 0.2 | 27.0× |
| COCO | 512.4 | 1.6 | 320.3× | 66.1 | 1.1 | 60.1× |
| NUS | 991.8 | 2.5 | 396.7× | 83.8 | 1.6 | 52.4× |
| Average | 398.3 | 1.1 | **362.1×** | 40.3 | 0.8 | **50.4×** |

## 5. Conclusion

We presented **PIAA**, a training-free framework that reformulates multi-label recognition with vision–language models as patch-level inference followed by adaptive aggregation. By combining Patch-based Visual Classifier Learning (PVCL) to reduce the patch-wise vision–language modality gap and Prediction Adaptive Aggregation (PAA) to consolidate sparse spatial evidence, **PIAA** delivers consistent gains across diverse multi-label benchmarks with minimal additional computation. **Limitations and future work.** Our approach still depends on the quality of the initial patch predictions (e.g., the reliability of patch embeddings and the purity of the selected patch bank). When early patch evidence is noisy or biased, the subsequent rectification and aggregation can be affected. An important future direction is to develop more robust patch acquisition and selection mechanisms, as well as reliability-aware calibration, to further improve stability under challenging clutter and co-occurrence.

## Acknowledgements

This work is supported by the Basic Research Project of Yunnan Province (Grant No. 202501CF070004), Xingdian Talent Support Program, and Intelligent Computing Center, Yunnan Normal University.

## Impact Statement

This paper presents work whose goal is to advance the field of Machine Learning. There are many potential societal consequences of our work, none which we feel must be specifically highlighted here.

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

# A. Appendix

## A.1. Qualitative Comparison of Activation Maps

Fig. 5 compares class activation maps. While the baseline suffers from severe contextual attention diffusion, PVCL successfully concentrates activations strictly on target objects. However, the bottom row reveals a limitation: extremely small targets (e.g., fork, remote) are easily overshadowed by dominant surrounding features due to the standard patch resolution limit. This causes activation diffusion, presenting a bottleneck for future work.

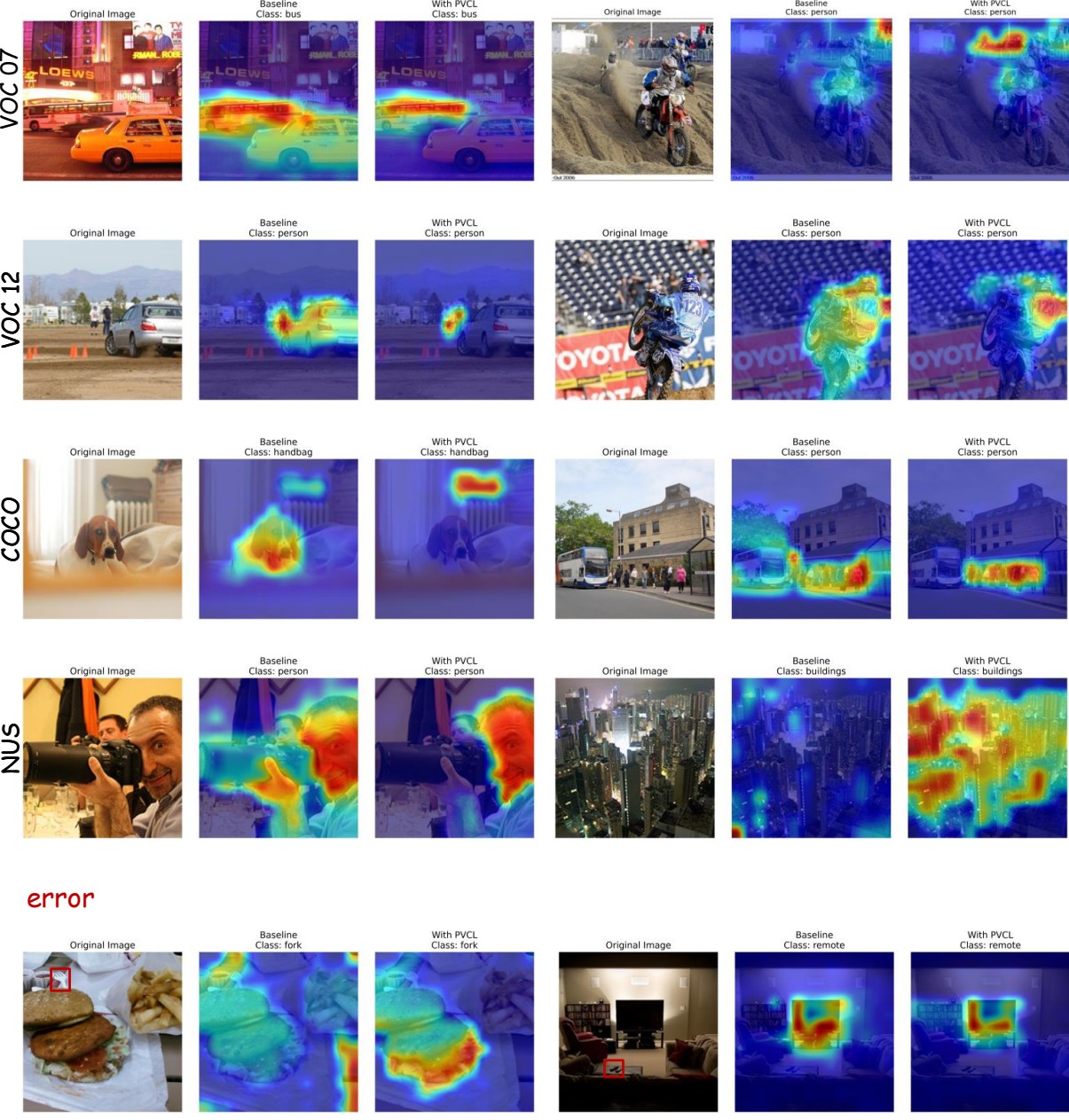

*Figure 5.* Comparison of class activation maps. PVCL tightly concentrates activations on target objects, correcting the baseline's contextual attention diffusion. The bottom row shows failure cases on extremely small targets due to patch-level resolution limits.

## A.2. Qualitative Visualization of Selected Patches

Fig. 6 visualizes the top-$K$ patches retained by our entropy-driven selection. This approach successfully isolates discriminative foregrounds (e.g., trains, leaves) while fading out uninformative backgrounds. However, the bottom row reveals occasional failures due to semantic co-occurrence bias, such as highlighting the rider instead of the motorbike, or the net instead of the soccer match. These ambiguous cases directly underscore the necessity of our subsequent PAA module, which leverages the global [CLS] anchor to regularize such localized semantic deviations.

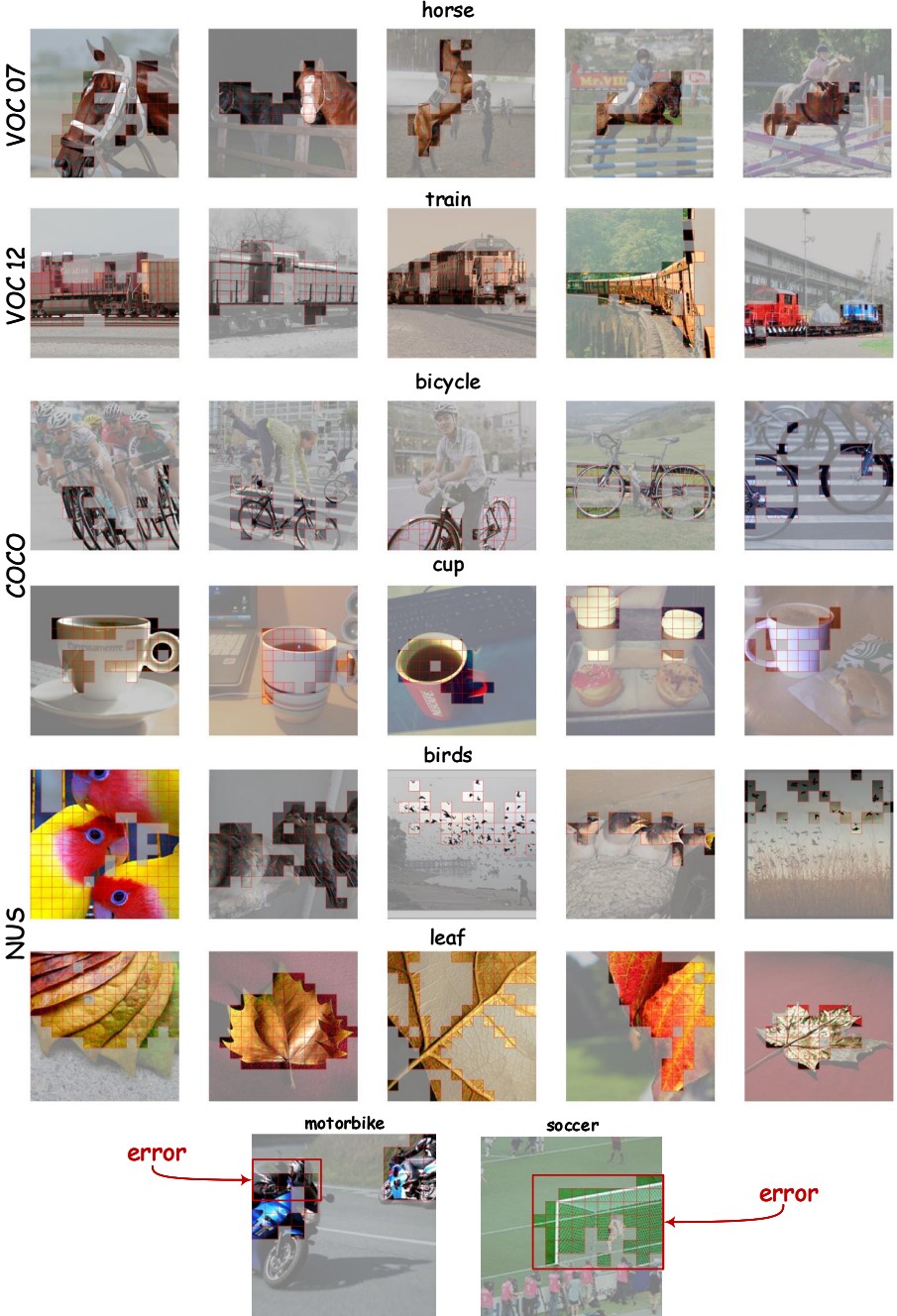

*Figure 6.* Qualitative visualization of the entropy-driven patch selection. Retained top-$K$ patches are displayed clearly, while discarded backgrounds are faded. The bottom row illustrates typical failure cases caused by semantic co-occurrence bias (e.g., motorbike, soccer).

### A.3. Empirical Analysis of Scale Complementarity

To empirically validate the motivation behind Patch-level Inference followed by Adaptive Aggregation (PIAA), Tab. 6 details the performance breakdown (mAP) on large versus small objects across the COCO, NUS, and VOC12 datasets.

The results reveal a clear scale-based divergence. Max-aggregated *Patch* scores excel at recognizing small, localized objects (e.g., *cup*, *bottle*, *mouse*), significantly outperforming the global [CLS] token. Conversely, the holistic [CLS] representation is notably more reliable for large, globally salient objects (e.g., *giraffe*, *aeroplane*) that typically span multiple patches.

By adaptively fusing both predictions, our PIAA strategy successfully exploits local spatial granularity alongside global semantic stability. As demonstrated in Tab. 6, this synergy effectively rescues small object recognition—providing substantial gains over the [CLS] baseline—while preserving and occasionally improving the strong baseline performance on large targets. This firmly confirms the fundamental complementarity between local patch evidence and global scene-level representations.

*Table 6.* Performance breakdown (mAP) on large vs. small objects across different datasets, demonstrating the clear complementarity between CLS and Patch predictions, and the ultimate effectiveness of our PIAA.

| | | **COCO** | | | | **NUS** | | | | **VOC12** | | |
|---|---|---|---|---|---|---|---|---|---|---|---|---|
| **Scale** | Class | CLS | Patch | PIAA | Class | CLS | Patch | PIAA | Class | CLS | Patch | PIAA |
| | giraffe | 99.3 | 95.5 | 99.4 | rainbow | 82.5 | 50.1 | 82.6 | aeroplane | 99.4 | 99.0 | 99.6 |
| Large | elephant | 98.4 | 96.8 | 98.8 | horses | 78.6 | 75.2 | 78.9 | horse | 97.3 | 96.6 | 98.4 |
| | zebra | 97.9 | 96.8 | 98.1 | sky | 72.4 | 61.3 | 72.4 | cow | 97.0 | 94.4 | 98.7 |
| | cup | 24.0 | 32.1 | 30.9 | soccer | 56.1 | 58.4 | 58.3 | tv monitor | 77.2 | 85.5 | 80.4 |
| Small | spoon | 17.3 | 31.6 | 23.0 | flags | 52.1 | 50.4 | 54.7 | bottle | 65.6 | 94.1 | 95.6 |
| | mouse | 11.2 | 51.0 | 16.1 | leaf | 17.1 | 21.6 | 18.7 | dining table | 63.6 | 75.7 | 72.9 |

