# OpenReview forum: "[CLS] is Not Enough: Multi-Label Recognition via Patch-Level Inference and Adaptive Aggregation"
_ICML.cc/2026/Conference — ICML 2026 regular_

### Official Review · Reviewer_cwBx · 2026-03-06

**Soundness:** 3
**Presentation:** 3
**Significance:** 3
**Originality:** 3
**Overall Recommendation:** 4
**Confidence:** 3

**Summary:**

This paper studies the problem of multi-label image recognition with CLIP, where using the global [CLS] token as the sole visual representation is insufficient to capture multiple co-existing objects. The authors propose a new framework to combine patch-level predictions into final multi-label outputs. The approach can be integrated with CLIP-style models. Extensive experiments demonstrate that the proposed method improves multi-label recognition performance over CLIP-based baselines.

**Compliance With Llm Reviewing Policy:**

Affirmed.

**Final Justification:**

This paper proposes PIAA, a training-free method for multi-label recognition. The motivation is clear that a single CLS token is insufficient to capture multiple targets. The combination of patch-level visual classification with adaptive aggregation is reasonably novel and well aligned with the problem setting, as patch-level features play an important role in multi-label scenarios. The method is clearly described, and the experimental results demonstrate strong performance across multiple benchmarks. During the rebuttal period, the authors provided additional qualitative results showing that the PVCL module effectively focuses on semantically relevant regions and also showing the failure cases. Regarding the sensitivity of top-k, the authors proposed an improved dynamic filtering strategy that reduces sensitivity. However, this improvement was introduced during the rebuttal stage and partially addresses the sensitivity issue, as the reported results still exhibit some fluctuations. Overall, considering the well-designed method and strong performance, I recommend a weak accept.

**Key Questions For Authors:**

1.	Is there an adaptive mechanism to determine the optimal k without manual tuning? If the method is highly sensitive to k, it significantly reduces the practical value.

2.	Can you provide visualizations demonstrating that the PVCL module focuses on semantically relevant regions compared to a baseline without this module?

3.	The proposed PVCL module relies on constructing an external "Patch Bank" by extracting features from a large number of unlabeled images, whereas other baselines do not. Does this comparison remain fair?

**Limitations:**

There is no discussion or qualitative analysis of failure cases.

**Strengths And Weaknesses:**

Strengths:

1.	The motivation is clear and the patch-level classification and adaptive aggregation components are clearly formulated.

2.	The paper presents a new combination of patch-level visual classification with adaptive aggregation specifically tailored for CLIP-based multi-label recognition.

3.	The method achieves strong results.

Weaknesses:

1.	Although the quantitative results are strong, the paper provides limited qualitative analysis.

2.	May lack comparisons with recent works, such as “More Reliable Pseudo-labels, Better Performance: A Generalized Approach to Single Positive Multi-label Learning” and “Correlative and Discriminative Label Grouping for Multi-Label Visual Prompt Tuning.”

3.	The parameter k varies drastically between datasets, more sensitivity analysis is needed to show how performance fluctuates with different values.

---

> ### Author Rebuttal · Authors · 2026-03-31
>
> We sincerely thank the reviewer for insightful comments, including constructive feedback, which helped us improve our work.
> >Q1. Although the quantitative results are strong, the paper provides limited qualitative analysis.
>
> A1.  Thank you for the valuable suggestion.  Following your suggestion as well as those of other reviewers, we will include more qualitative analysis in the revised version.
> ___
> >Q2. May lack comparisons with recent works, such as “More Reliable Pseudo-labels, Better Performance: A Generalized Approach to Single Positive Multi-label Learning” and “Correlative and Discriminative Label Grouping for Multi-Label Visual Prompt Tuning.”
>
> A2. Thank you for the helpful suggestion. Regarding the first paper,  we have actually included it in our comparison. It is listed as **AEVLP (Tran et al., 2025)** in Table 1 under the weakly supervised setting, where our training-free PIAA still achieves highly competitive performance. Regarding the second paper, it is a supervised fine-tuning-based method and was therefore not included in our original comparison, which mainly focuses on training-free, unsupervised, and weakly supervised settings. Nevertheless, following your suggestion, we will include a clearer discussion of more related recent works in the revised version and better explain the scope of our comparisons.
> ___
> >Q3. The parameter k varies drastically between datasets; more sensitivity analysis is needed to show how performance fluctuates with different values.
>
> A3. Thank you for this valuable comment. We agree that the sensitivity of $K$ is not uniform across datasets. As shown in Fig. 4, the performance is relatively stable on VOC2007 and VOC2012, while it is noticeably more sensitive on COCO and NUS-WIDE. One possible reason is that the current fixed Top-$K$ selection does not explicitly account for dataset complexity or class-wise variability. In more challenging datasets such as COCO and NUS-WIDE, different classes may differ substantially in both the number of reliable patches and the quality of their predictions. As a result, using the same fixed $K$ for all classes may lead to suboptimal patch bank construction and larger performance fluctuations.
>
> Motivated by this observation, and consistent with our response above, we further revise the memory construction strategy by introducing an additional class-wise filtering step after the initial fixed-$K$ selection. Specifically, after selecting high-confidence patches following the original design, we apply a class-wise threshold defined by the mean and standard deviation of prediction scores to remove unreliable patches, and then relearn the classifier from the refined patch bank. As shown in **Table 1**, this modification improves the stability of the model and reduces its sensitivity to the choice of $K$, without introducing additional hyperparameters.
>
> Table 1: Sensitivity analysis of the initial parameter $K$ using our revised dynamic filtering mechanism.
> | $K$ | VOC12 | **VOC07** | COCO | NUS  |
> | ----- | ----- | --------- | ---- | ---- |
> | 32    | 88.2  | 88.9      | 67.3 | 44.9 |
> | 64    | 88.6  | 89.3      | 72.2 | 48.2 |
> | 128   | 89.2  | 89.9      | 73.1 | 50.0   |
> | 512   | 92.2  | 92.5      | 73.2 | 50.2 |
> | 1024  | 92.2  | 92.5      | 72.3 | 49.8 |
> | 2048  | 91.6  | 91.7      | 71.8 | 48.7 |
> ___
> >Q4. Is there an adaptive mechanism to determine the optimal k without manual tuning? If the method is highly sensitive to k, it significantly reduces the practical value.
>
> A4. See A3.
> ___
> >Q5. Can you provide visualizations demonstrating that the PVCL module focuses on semantically relevant regions compared to a baseline without this module?
>
> A5. Yes. We have provided side-by-side visual comparisons here: https://anonymous.4open.science/r/test-EF0E/pvcl.jpg. The heatmaps clearly demonstrate that without PVCL, attention is often scattered and distracted by background noise. In contrast, integrating PVCL explicitly bridges the modality gap, enforcing the visual classifier to focus precisely on semantically relevant local foreground regions. We will include these in the revised supplementary material.
> ___
> >Q5. The proposed PVCL module relies on constructing an external "Patch Bank" by extracting features from a large number of unlabeled images, whereas other baselines do not. Does this comparison remain fair?
>
> A5. Thank you for the comment. We note that this setting follows the common unsupervised paradigm where unlabeled data is allowed. Methods such as CDUL leverage unlabeled data to achieve stable generalization, but require substantial training and computational cost. In contrast, inference-only methods like TagCLIP avoid such cost but cannot exploit information from unlabeled data. Our method provides a middle ground: it leverages unlabeled data to improve adaptation, while remaining training-free and computationally efficient.

---

> > ### Author Rebuttal · Reviewer_cwBx · 2026-04-03
> >
> > I would like to thank the authors for their additional experiments and clarifications. Although the revised dynamic filtering mechanism partially addresses the sensitivity problem,  it is newly proposed in the rebuttal period. However, as my major concerns have been addressed, I will raise my score to Weak Accept. I highly recommend that the authors incorporate all the promised revisions into the final version.

---

> > > ### Author Response · Authors · 2026-04-04
> > >
> > > Thank you for reading our rebuttal and updating your score. **We will carefully incorporate all the promised revisions into the final version** to further improve the quality of the paper.

---

### Official Review · Reviewer_p6qp · 2026-03-07

**Soundness:** 4
**Presentation:** 2
**Significance:** 3
**Originality:** 3
**Overall Recommendation:** 4
**Confidence:** 3

**Summary:**

This paper proposes a training-free multi-label recognition framework called PIAA. Its core argument is that relying solely on CLIP’s global [CLS] representation is insufficient for recognizing multiple objects co-existing in a single image, and it therefore reformulates the task as “patch-level inference + adaptive aggregation.”

(a). It leverages segmentation-style patch representations and introduces unsupervised Patch-based Visual Classifier Learning (PVCL) to bridge the modality gap between patch features and text prototypes, thereby improving the reliability of local classification.

(b) It uses a Prediction Adaptive Aggregation (PAA) module to fuse patch-level category scores with the global [CLS] prediction, suppressing noise while preserving local object evidence.

**Compliance With Llm Reviewing Policy:**

Affirmed.

**Final Justification:**

My concerns have been addressed.

**Key Questions For Authors:**

See weaknesses.

**Limitations:**

yes.

**Strengths And Weaknesses:**

**Strengths:**
1. By learning an unsupervised visual classifier through PVCL, the method makes the matching between patch features and textual category prototypes more reliable.
2. The paper points out that relying solely on the global [CLS] feature tends to overlook small objects, local objects, or multiple co-existing targets in an image, whereas patch-level inference can capture different semantic regions in a more fine-grained manner, making it more suitable for multi-label recognition.

**Weaknesses:**
1. If the initial CLIP model itself suffers from severe semantic confusion and a modality gap when processing patches, then the selected “high-confidence” patch bank may already be biased. In the subsequent GDA stage, this bias could be further amplified, causing the model to overfit to the erroneous predictions of the original CLIP. The paper lacks a discussion of the risk of such error propagation.
2. In Table 2, the performance improves substantially when PVCL and PAA are used together. This is an interesting phenomenon, and I hope the authors can provide further discussion and analysis.
3. The setting $\alpha = 0.9$ implies that the [CLS] token contributes only $10\%$. Please provide an ablation study on $\alpha$ to examine the degree of influence from [CLS]. Moreover, the claim that the same parameter setting, $\alpha = 0.9$, achieves the best performance across all datasets is somewhat questionable. This suggests that the model makes very limited use of the global [CLS] information, contributing only 10%. Logically, this also raises a reverse question: if the global information is so unimportant, why is it still used as the “anchor” for aggregation?
4. In Section 3.4, in what sense is the method adaptive?
5. Why is the score of TagCLIP on NUS-WIDE only 38.7, which seems unusually low? Equation (6) is the expression of entropy, but in Equation (7), Top-K appears to select samples with high entropy and low confidence. However, line 5 of Algorithm 1 again uses $\min H(z_i)$.
6. The PVCL module uses a large amount of unlabeled data to learn the visual classifier, which does not seem entirely fair.

---

> ### Author Rebuttal · Authors · 2026-03-31
>
> We sincerely thank the reviewer for insightful comments, including constructive feedback, which helped us improve our work.
> >Q1. If the initial CLIP model itself suffers from severe semantic confusion and a modality gap when processing patches, then the selected “high-confidence” patch bank may already be biased,...
>
> A1. Thank you for the insightful comment. We agree that biased patch selection may lead to error propagation, and indeed, a stronger segmentation model provides a better starting point for GDA, as reflected in Table 3. However, we also observe that PIAA brings larger gains for weaker baselines. For example, although SCLIP and SC-CLIP exhibit a significant performance gap before applying PIAA, this gap becomes much smaller after incorporating PIAA. This suggests that PIAA does not simply amplify the bias of the initial model, but instead helps mitigate it to some extent.
>
> ___
> >Q2. In Table 2, the performance improves substantially when PVCL and PAA are used together. This is an interesting phenomenon, and I hope the authors can provide further discussion and analysis.
>
> A2. The main reason is that the two modules address different bottlenecks in patch-based multi-label recognition. Specifically, PVCL learns a patch-based visual classifier to reduce the modality gap and improve the reliability of patch-level predictions, whereas PAA focuses on aggregating such local evidence into image-level outputs by adaptively combining $S_{\text{patch}}$ and $S_{\text{cls}}$. As discussed in A4 of reviewer " cBF7", these two signals are clearly complementary: $S_{\text{patch}}$ is more suitable for recognizing small or localized objects, while $S_{\text{cls}}$ is more reliable for large or globally salient objects. Therefore, their combination effectively exploits both local sensitivity and global semantic stability, leading to substantially larger gains than using either component alone.
> ___
> >Q3. The setting $\alpha=0.9$ implies that the [CLS] token contributes only,...
>
> A3. As discussed in A2, both patch-level predictions and the CLS-based prediction are important for the final result. Patch-level predictions are crucial for improving the recognition of small or less salient objects, but they are more sensitive to outliers. In contrast, the CLS token provides reliable global semantic information, which helps maintain the accuracy of dominant objects and mitigates the effect of noisy patch activations. Therefore, even though the CLS contribution is relatively small (e.g., 10\%), it plays an essential role as a global anchor to regularize patch-level predictions and suppress outliers.
> ___
> >Q4. In Section 3.4, in what sense is the method adaptive?
>
> A4. The “adaptive” nature mainly lies in how patch-level and global predictions are combined. Specifically, the method adaptively aggregates category-wise patch responses by selecting the most confident evidence (via max aggregation) and further balancing it with the CLS-based prediction. This allows the model to dynamically emphasize reliable patch-level signals while using global information to suppress outliers.
> ___
> >Q5. Why is the score of TagCLIP on NUS-WIDE only 38.7, which seems unusually low?
>
> A5. Thank you for pointing this out. We would like to clarify that the value 38.7 in Table~1 is our reproduced result under the unified evaluation protocol used in this paper, rather than an official number reported in the original TagCLIP paper or its public repository. In fact, the original TagCLIP does not provide an official NUS-WIDE result. A possible reason for its relatively low performance on NUS-WIDE is that TagCLIP mainly relies on local patch reasoning and refinement to generate image-level predictions. While this design is effective on datasets with relatively clear object-centric cues, it may be less suitable for NUS-WIDE, where label semantics are more diverse and many categories are associated with complex scene context or weakly localized visual evidence. Under such cases, the reliability of its local refinement pipeline can be weakened.
> ___
> >Q6. Equation (6) is the expression of entropy, but in Equation (7), Top-K appears to select samples with high entropy and low confidence. However, line 5 of Algorithm 1 again uses
> $\min H(z)$.
>
> A6. Thank you for pointing this out. This is a writing error, and we will correct it in the final version.
> ___
> >Q7. The PVCL module uses a large amount of unlabeled data to learn the visual classifier, which does not seem entirely fair.
>
> A7. This setting follows the common unsupervised paradigm where unlabeled data is allowed. Methods such as CDUL leverage unlabeled data to achieve stable generalization, but require substantial training and computational cost. In contrast, inference-only methods like TagCLIP avoid such cost but cannot exploit information from unlabeled data. Our method provides a middle ground: it leverages unlabeled data to improve adaptation, while remaining training-free and computationally efficient.

---

> > ### Author Rebuttal · Reviewer_p6qp · 2026-04-03
> >
> > Thank you to the authors for the response. I will maintain my score.

---

> > > ### Author Response · Authors · 2026-04-03
> > >
> > > **We sincerely thank the reviewer for positively recognizing the contributions of our work.** We will carefully incorporate all the clarifications and additional analyses into the revised manuscript to further improve the quality of the paper.

---

### Official Review · Reviewer_cBF7 · 2026-03-10

**Soundness:** 4
**Presentation:** 3
**Significance:** 3
**Originality:** 4
**Overall Recommendation:** 5
**Confidence:** 5

**Summary:**

This paper proposes a training-free framework for multi-label recognition with VLM. Instead of relying only on the global [CLS] token, the method performs patch-level inference and then adaptively aggregates patch and global predictions. The paper shows strong results on several benchmarks, with clear gains in both accuracy and efficiency.

**Compliance With Llm Reviewing Policy:**

Affirmed.

**Final Justification:**

The authors have addressed  my concerns.

**Key Questions For Authors:**

1. Using segmentation-style or patch-level reasoning for multi-label recognition seems quite natural. The authors should more clearly clarify the distinction between the proposed method and prior open-vocabulary semantic segmentation approaches.
2. Will using the [CLS] token in the aggregation stage (Eq. 13) introduce distracting or irrelevant information?
3. How does the classifier learned via GDA remain aligned with the textual semantics？
4. Can the proposed method be directly extended to semantic segmentation？
5. The Overall Results section should not only emphasize the superiority of the proposed method, but also include more analysis of the results.
Overall, I think this work is meaningful for the multi-label recognition community. If the authors can clearly address the concerns above, I would be happy to further increase my rating.

**Limitations:**

Yes

**Strengths And Weaknesses:**

Strengths:
1. The paper is clearly written and generally easy to follow.
2. The proposed training-free framework is appealing.
3. The proposed method shows clear advantages in both experimental performance and computational efficiency.

Weaknesses:
1. Although the empirical results are strong, the technical novelty appears somewhat limited. In particular, learning a classifier with GDA or similar analytical schemes has already been used in prior zero-shot or training-free classification works. The paper should better explain what the main methodological novelty is.
2. The efficiency gains are very impressive, but the paper does not provide enough analysis of why the proposed method is so much faster than the compared training-free methods.

---

> ### Author Rebuttal · Authors · 2026-03-31
>
> >Q1. Although the empirical results are strong, the technical novelty appears somewhat limited. In particular, learning a classifier with GDA or similar analytical schemes has already been used in prior zero-shot or training-free classification works.
>
> A1. Thank you for the comment. We agree that GDA itself is not novel and can be replaced by other analytical classifier learning methods. Our contribution lies in showing that learning a visual classifier to reduce the vision–language modality gap is an effective way to improve patch-level predictions. We further propose a reliable patch selection strategy for classifier learning and an adaptive aggregation mechanism for robust image-level prediction. These components together distinguish our method from prior work.
> ___
> >Q2. The efficiency gains are very impressive, but the paper does not provide enough analysis of why the proposed method is so much faster than the compared training-free methods.
>
> A2. Thank you for the comment. The efficiency gains mainly come from avoiding both training and repeated inference. Compared to training-based methods, our approach eliminates the costly optimization process. Compared to existing training-free methods, our method only requires a single forward pass, without multi-crop or iterative inference. This avoids repeated image encoding and significantly reduces computational overhead.
> ___
> >Q3. Using segmentation-style or patch-level reasoning for multi-label recognition seems quite natural. The authors should more clearly clarify the distinction between the proposed method and prior open-vocabulary semantic segmentation approaches.
>
> A3. Thank you for the comment. The key difference is that existing open-vocabulary segmentation methods mainly focus on improving patch feature quality, while largely overlooking the vision–language modality gap. In contrast, we explicitly address this gap by learning a visual classifier for patch-level prediction. In addition, we show that how to adaptively aggregate patch-level predictions into reliable image-level outputs is another critical yet underexplored problem, which is addressed by our method.
>
> ___
> >Q4. Will using the [CLS] token in the aggregation stage (Eq. 13) introduce distracting or irrelevant information?
>
> A4. We appreciate the insightful question.  As shown in **Table 1**, $S_{\text{patch}}$ and $S_{\text{cls}}$ are clearly complementary: $S_{\text{patch}}$ is more suitable for recognizing small or localized objects, while $S_{\text{cls}}$ is more reliable for large or globally salient objects. Therefore, their combination effectively exploits both local sensitivity and global semantic stability, leading to substantially larger gains than using either component alone.
>
>
> Table 1: Performance breakdown (mAP) on large-scale vs. small objects across different datasets.
> |       | COCO     |      |       |      | **NUS** |      |       |      | **VOC12**    |      |       |      |
> | ----- | -------- | ---- | ----- | ---- | ------- | ---- | ----- | ---- | ------------ | ---- | ----- | ---- |
> | Scale | Class    | CLS  | Patch | PIAA | Class   | CLS  | Patch | PIAA | Class        | CLS  | Patch | PIAA |
> | Large | giraffe  | 99.3 | 95.5  | 99.4 | rainbow | 82.5 | 50.1  | 82.6 | aeroplane    | 99.4 | 99.0    | 99.6 |
> |       | elephant | 98.4 | 96.8  | 98.8 | horses  | 78.6 | 75.2  | 78.9 | horse        | 97.3 | 96.6  | 98.4 |
> |       | zebra    | 97.9 | 96.8  | 98.1 | sky     | 72.4 | 61.3  | 72.4 | cow          | 97.0   | 94.4  | 98.7 |
> | Small | cup      | 24.0   | 32.1  | 30.9 | soccer  | 56.1 | 58.4  | 58.3 | tv monitor   | 77.2 | 85.5  | 80.4 |
> |       | spoon    | 17.3 | 31.6  | 23.0   | flags   | 52.1 | 50.4  | 54.7 | bottle       | 65.6 | 94.1  | 95.6 |
> |       | mouse    | 11.2 | 51.0    | 16.1 | leaf    | 17.1 | 21.6  | 18.7 | dining table | 63.6 | 75.7  | 72.9 |
>
>
> ___
> >Q5. How does the classifier learned via GDA remain aligned with the textual semantics?
>
> A5. The alignment is preserved by incorporating the textual prior during GDA estimation. Specifically, as shown in Eq. (8) and Eq. (9), the class-wise prototypes are constructed using soft assignments derived from the language classifier. This injects semantic information into the visual classifier, ensuring that the learned GDA classifier remains aligned with textual semantics.
> ___
> >Q6. Can the proposed method be directly extended to semantic segmentation?
>
> A6. Yes.  In particular, PVCL enhances localized patch predictions by reducing the vision--language modality gap, which is also crucial for dense prediction tasks.

---

> > ### Author Rebuttal · Reviewer_cBF7 · 2026-04-01
> >
> > My concerns are generally addressed.

---

> > > ### Author Response · Authors · 2026-04-03
> > >
> > > **Thank you for reading our rebuttal and updating your score.** We will carefully incorporate all the clarifications and additional analyses into the revised manuscript to further improve the quality of the paper.

---

### Official Review · Reviewer_J6JJ · 2026-03-12

**Soundness:** 4
**Presentation:** 3
**Significance:** 3
**Originality:** 3
**Overall Recommendation:** 4
**Confidence:** 3

**Summary:**

This paper proposes PIAA, a training-free framework for multi-label recognition that addresses the limitation of merely relying on the global [CLS] token bottleneck in CLIP by performing patch-level inference followed by adaptive aggregation. The method consists of two key components: 1) Patch-based Visual Classifier Learning (PVCL), which bridges the vision–language modality gap by learning an unsupervised visual classifier via Gaussian Discriminant Analysis on the patch banks, and 2) Prediction Adaptive Aggregation (PAA), which fuses max-pooled patch-level scores with the global [CLS] prediction through a convex combination. Experiments on four multi-label benchmarks (VOC2007, VOC2012, MS-COCO, NUS-WIDE) demonstrate consistent improvements over existing training-free and unsupervised baselines, with notable efficiency gains.

**Compliance With Llm Reviewing Policy:**

Affirmed.

**Final Justification:**

As most of my concerns are addressed, I have raised my score.

**Key Questions For Authors:**

1. Could you provide results with genuinely different VLM backbones (e.g., SigLIP, ViT-L/14) to demonstrate the generality of PIAA beyond CLIP ViT-B/16?

2. What explains the main difference in hyperparameter sensitivity between VOC and COCO/NUS-WIDE?

3. Can you provide qualitative examples of the patches selected for the patch bank, showing whether they reliably correspond to foreground objects across different classes and datasets?

**Limitations:**

yes

**Strengths And Weaknesses:**

Strengths
1. The paper is well-written and easy to follow. The motivation is clearly articulated—namely, that the single [CLS] token is insufficient for multi-label scenarios—and the logical flow from problem identification to the proposed solution is coherent throughout.
2. The motivation that the single [CLS] token is insufficient for the multi-label recognition task is clear. Furthermore, the proposed method addressing the challenge of modality gap is reasonable.
3. The proposed method is training-free, which requires no gradient updates or parameter fine-tuning. This leads to a significant practical advantage.
4. Through extensive experiments on four benchmarks, the authors validate the effectiveness of the proposed method. Furthermore, the ablation study systematically isolates the contributions of each component (disentanglement, PVCL, PAA), demonstrating their individual and synergistic effects.

Weakness
1. Several aspects of the experimental setup lack clarity:
(a) The description of baseline methods is insufficient. For readers unfamiliar with all cited works, it is difficult to understand what each baseline does and how it differs from PIAA without referring to external papers.
(b) Table 3 is framed as evaluating PIAA across various backbones, but the variants (SCLIP, ITACLIP, SC-CLIP) are not truly different backbones. They are different segmentation-style attention modification strategies applied on the same CLIP ViT-B/16. This framing is misleading and overstates the generalizability of the approach.
(c) The main results in Table 1 use SC-CLIP as the segmentation method, which itself already provides strong performance (e.g., 68.8 mAP on COCO). It is unclear whether the comparison with baselines that do not use such disentanglement is fair. The authors should clarify how much of the gain is attributable to PIAA's own contributions versus the underlying segmentation method.
(d) To convincingly demonstrate generalization, experiments on different VLM backbones (e.g., SigLIP, EVA-CLIP, or different ViT scales such as ViT-L/14) are needed. Without this, the claim of a general framework remains insufficiently supported.

2. The authors should conduct further analysis of hyperparameter sensitivity across datasets. Figure 4 shows that performance on VOC2007 and VOC2012 is remarkably stable across different values of both $K$ and $\alpha$, while COCO and NUS-WIDE exhibit more pronounced sensitivity. The paper does not adequately explain this discrepancy. Without this analysis, there is a concern that the method's effectiveness may be dataset-dependent.

3. There is a lack of qualitative analysis on patch selection. The paper would benefit significantly from qualitative visualizations showing which patches are selected into the patch bank. Specifically, it is important to verify whether the Top-K high-confidence patches genuinely correspond to salient foreground objects or whether they inadvertently include background or co-occurring context patches. Figure 1 provides attention/activation map comparisons, but this does not directly address the patch bank curation process. Visualizing the selected patches would strengthen the paper's claims about the reliability of the PVCL module and provide insight into failure cases.

---

> ### Author Rebuttal · Authors · 2026-03-31
>
> We sincerely thank the reviewer for insightful comments, including constructive feedback, allowing us to improve our work.
> >Q1. The description of baseline methods is insufficient.
>
> A1. We agree that the baseline descriptions can be clearer. In the revision, we will add summaries of each baseline and explicitly highlight their key differences from PIAA.
>
> ___
> >Q2. Table 3 is framed as evaluating PIAA across various backbones, but the variants are not truly different backbones.
>
> A2. We have conducted additional experiments with different backbones. As shown in **Table 1**, PIAA consistently brings improvements. We will further supplement these results and revise the presentation to better reflect the generalizability of our approach.
>
> Table 1: PIAA evaluated on genuinely different VLM backbones and ViT scales (mAP).
> |        Backbone       |  Method  | VOC2012 | VOC2007 | COCO | NUS-WIDE |  AVG |
> | :-------------------: | :------: | :-----: | :-----: | :--: | :------: | :--: |
> |    CLIP (ViT-L/14)    | Baseline |   70.5  |   69.9  | 45.8 |   32.6   | 54.7 |
> |                       |   +PIAA  |   88.7  |   89.1  | 68.8 |   46.7   | 73.3 |
> | EVA02-CLIP (ViT-B/16) | Baseline |   83.4  |    84.0   | 60.5 |   38.3   | 66.6 |
> |                       |   +PIAA  |   91.6  |   92.3  | 72.4 |   48.6   | 76.2 |
> | EVA02-CLIP (ViT-L/14) | Baseline |   85.5  |   83.9  | 73.3 |    39.0    | 70.4 |
> |                       |   +PIAA  |   93.7  |   93.6  | 80.1 |   49.8   | 79.3 |
> ___
> >Q3. The main results in Table 1 use SC-CLIP as the segmentation method, which itself already provides strong performance (e.g., 68.8 mAP on COCO). It is unclear whether the comparison with baselines that do not use such disentanglement is fair. The authors should clarify how much of the gain is attributable to PIAA's own contributions versus the underlying segmentation method.
>
> A3. We clarify that Table 3 (in the paper) and **Table 1** (in our response) evaluate PIAA on top of different segmentation methods, where consistent improvements are observed across all variants. More importantly, we observe that PIAA brings larger gains for weaker baselines. For example, although SCLIP and SC-CLIP exhibit a significant performance gap before applying PIAA, this gap becomes much smaller after incorporating PIAA. This indicates that the performance improvements are primarily attributed to PIAA itself rather than the underlying segmentation method.
> ___
> >Q4. The authors should conduct further analysis of hyperparameter sensitivity across datasets. Figure 4 shows that performance on VOC2007 and VOC2012 is remarkably stable across different values of both and, while COCO and NUS-WIDE exhibit more pronounced sensitivity.
>
> A4. Thank you for this insightful comment. We agree that the higher sensitivity on more complex datasets, such as COCO and NUS-WIDE. One possible reason is that the current Top-$K$ selection strategy does not explicitly account for dataset complexity or class-wise variability. In more challenging datasets, different classes may differ significantly in both the number of reliable patches and the quality of their predictions.  As a result, using a fixed Top-$K$ for all classes may lead to suboptimal patch bank construction.
>
> Motivated by this observation, as well as the suggestions from other reviewers, we revise the memory construction strategy as follows.
> - First, we retain the original entropy-based fixed-$K$ selection to obtain an initial set of high-confidence patches, which is consistent with the original design.
> - Second, based on the learned classifier, we re-evaluate the patch bank and apply a class-wise threshold, defined by the mean and standard deviation of prediction scores, to filter out unreliable patches.
> - Finally, We relearn the classifier from the refined patch bank.
>
> In this way, the revised strategy better accounts for class-wise variability without introducing additional hyperparameters. As shown in **Table 2**, this modification improves the stability of the model.
>
> Table 2: Sensitivity analysis of the $K$ using our revised dynamic filtering mechanism.
> | $K$  | VOC12 | **VOC07** | COCO | NUS  |
> | ---- | ----- | --------- | ---- | ---- |
> | 32   | 88.2  | 88.9      | 67.3 | 44.9 |
> | 64   | 88.6  | 89.3      | 72.2 | 48.2 |
> | 128  | 89.2  | 89.9      | 73.1 | 50.0   |
> | 512  | 92.2  | 92.5      | 73.2 | 50.2 |
> | 1024 | 92.2  | 92.5      | 72.3 | 49.8 |
> | 2048 | 91.6  | 91.7      | 71.8 | 48.7 |
> ___
> >Q5. There is a lack of qualitative analysis on patch selection. The paper would benefit significantly from qualitative visualizations showing which patches are selected into the patch bank.
>
> A5. We have provided qualitative visualizations of the selected patches via an anonymous link: https://anonymous.4open.science/r/test-EF0E/top-k_select.jpg. The results clearly demonstrate that our Top-K selection strategy captures salient foreground objects. We will add these qualitative analyses to the revised supplementary material.

---

> > ### Author Rebuttal · Reviewer_J6JJ · 2026-04-03
> >
> > I sincerely appreciate the authors' detailed response, particularly the additional experiments on different backbones, varying $K$, and the qualitative analysis. As these address most of my concerns, I have raised my score.

---

> > > ### Author Response · Authors · 2026-04-04
> > >
> > > Thank you for reading our rebuttal and updating your score. **We will carefully incorporate all the promised revisions into the final version** to further improve the quality of the paper.

---

### Decision · Program_Chairs · 2026-04-30

**Decision:**

Accept (regular)

**Comment:**

The final reviewer scores for this paper are 4, 5, 4, and 4. The paper’s main strengths are its clear motivation, practical training-free design, and strong empirical results. Reviewers also found the paper generally well written and easy to follow, with convincing performance and efficiency advantages across multiple benchmarks.

However, the paper lacks sufficient discussion of recent related work, and the authors are encouraged to include a more complete comparison and positioning of recent literature in the camera-ready version. Overall, all reviewers were positive about the paper. Given the generally favorable reviews, I recommend acceptance.